# Duration-Dependent Lung Injury Induced by High-Intensity Electric Field Exposure: Histopathological and Immunoinflammatory Insights

**DOI:** 10.3390/ijms262010125

**Published:** 2025-10-17

**Authors:** Süleyman Emre Akın, Orhan İmeci, Halil Aşcı, Arzu Ulusoy, Adem Milletsever, Özlem Özmen, Selçuk Çömlekçi

**Affiliations:** 1Department of Thoracic Surgery, Faculty of Medicine, Suleyman Demirel University, Isparta 32260, Türkiye; 2Department of Pharmacology, Faculty of Medicine, Suleyman Demirel University, Isparta 32260, Türkiye; orhanimeci@sdu.edu.tr (O.İ.); halilasci@sdu.edu.tr (H.A.); 3Department of Bioengineering, Institute of Natural and Applied Sciences, Suleyman Demirel University, Isparta 32260, Türkiye; arzuulusoy32@gmail.com; 4Department of Pathology, Faculty of Veterinary Medicine, Burdur Mehmet Akif Ersoy University, Burdur 15030, Türkiye; adem.milletsever@gmail.com (A.M.); ozlemozmen70@gmail.com (Ö.Ö.); 5Department of Electronics and Communication Engineering, Faculty of Engineering, Suleyman Demirel University, Isparta 32260, Türkiye; selcukcomlekci@sdu.edu.tr

**Keywords:** electric field, lung injury, acute-phase reactants, thoracic surgery, pulmonary inflammation

## Abstract

Patients undergoing thoracic surgery or intensive care are frequently exposed to high-voltage electric fields generated by medical devices; however, the duration-dependent effects of such exposure on lung tissue remain unclear. This study aimed to investigate the histopathological and immunoinflammatory effects of exposure to a uniform 10 kV/m electric field for varying durations using a rat model. Thirty-five adult female Wistar rats were randomly assigned to five groups (*n* = 7): control, and 1, 5, 15, and 30 min exposure groups. Lung tissues were analyzed histologically with hematoxylin and eosin staining, and the immunohistochemical expression of IL-1β, RANKL, and TNF-α was semi-quantitatively assessed. Histopathological examination revealed a duration-dependent increase in lung injury, with the 30 min group showing marked epithelial loss, mononuclear infiltration, edema, and vascular congestion (*p* < 0.001). The expression of IL-1β, RANKL, and TNF-α remained minimal in the 1–15 min groups but was significantly elevated in the 30 min group (*p* < 0.001). These findings suggest that prolonged exposure to high-voltage electric fields induces substantial pulmonary inflammation and tissue damage, indicating the presence of a threshold beyond which inflammatory pathways are abruptly activated. These results highlight the importance of establishing safety guidelines for electric field exposure in clinical settings.

## 1. Introduction

In modern thoracic surgery and critical care environments, the integration of high-voltage electrical devices has become increasingly ubiquitous [1]. From electrocautery units and harmonic scalpels to intraoperative navigation systems and defibrillators, patients undergoing pulmonary surgery are routinely exposed to transient but intense electric fields [2]. Furthermore, in postoperative settings, particularly in intensive care units (ICUs) and mechanical ventilation contexts, prolonged exposure to stray electric fields emitted by surrounding medical equipment, including ventilators, monitors, and high-frequency oscillators, is inevitable [3]. While these technologies are indispensable to contemporary thoracic practice, the biological implications of electric field exposure on pulmonary tissue remain poorly understood, especially in relation to exposure duration and inflammatory responses [1,4].

Emerging preclinical data suggests that electric fields can modulate cell membrane polarization, ion channel activity, and intracellular signaling cascades, particularly in electrically sensitive tissues such as the heart, nervous system, and respiratory epithelium [5]. However, investigations into how high-voltage electric fields influence lung parenchyma at the cellular and molecular level are notably scarce [1]. This represents a critical knowledge gap for thoracic surgeons, as the lung’s delicate alveolar–capillary interface is particularly susceptible to oxidative and inflammatory injury. Understanding whether operative or perioperative electric field exposure might inadvertently initiate pulmonary damage or exacerbate subclinical inflammation is essential for optimizing patient safety and postoperative outcomes.

In this context, the present study sought to characterize the histopathological and immunoinflammatory alterations in lung tissue following graded-duration exposure to a 10 kV/m electric field, using a controlled in vivo rat model that mimics perioperative exposure scenarios. Special emphasis was placed on the expression of key acute-phase inflammatory mediators, namely interleukin-1 beta (IL-1β), tumor necrosis factor-alpha (TNF-α), and receptor activator of nuclear factor kappa-B ligand (RANKL). These biomarkers were selected due to their well-established roles in early-phase inflammatory signaling, tissue remodeling, and immune cell recruitment in acute pulmonary injury [6,7].

IL-1β and TNF-α are canonical cytokines released rapidly in response to epithelial or endothelial stress, orchestrating neutrophil infiltration, vascular permeability, and cytokine amplification within the acute phase of lung inflammation [6,7]. RANKL, although traditionally associated with osteoimmunology, has been increasingly implicated in pulmonary inflammatory diseases, including acute lung injury and ventilator-associated inflammation, where it contributes to epithelial–mesenchymal transition and immune activation [7]. All three molecules are recognized as early-response mediators that may serve as sensitive indicators of electric field-induced tissue perturbation.

This study hypothesized that electric field exposure of increasing duration would induce a progressive inflammatory response in lung tissue, culminating in histopathological damage and cytokine upregulation consistent with acute-phase activation. By integrating histological assessment with immunohistochemical quantification of these markers, we aimed to elucidate whether the electric fields routinely encountered in thoracic surgical and ICU environments pose a risk of subclinical or overt lung injury. The results may offer crucial insights into safety thresholds for intraoperative and perioperative exposure, informing both device design and clinical protocols for vulnerable patient populations.

The intensity of 10 kV/m was selected based on prior standardization experiments in pressure injury and hepatic exposure models using the same parallel-plate configuration [8]. In this study, this level represented the highest non-thermal, biologically safe threshold capable of inducing measurable cellular responses without causing overt tissue necrosis. Exposure durations of 1, 5, 15, and 30 min were chosen to simulate clinically relevant scenarios ranging from short intraoperative bursts to prolonged intensive-care exposure intervals.

Our goal was to choose a daily exposure limit. According to WHO guidelines, the defined public limit is 10 kV/m in the power frequency range [9].

## 2. Results

### 2.1. Histopathological Findings

Histological examination of lung tissues across experimental groups revealed a clear, exposure duration-dependent progression in structural damage following high-voltage electric field exposure. In the control group, the alveolar spaces, bronchiolar epithelium, and interstitial areas maintained normal morphology with no observable pathological alterations. Rats exposed to a 1 min electric field exhibited only mild hyperemia, with negligible signs of tissue disruption. Notably, the 5 min exposure group demonstrated localized mononuclear cell infiltration and modest epithelial detachment in some regions, yet the overall parenchymal integrity remained largely preserved.

Progressive pathological changes became more apparent in the 15 min group, which showed focal accumulation of inflammatory cells within the peribronchiolar and perivascular regions, alongside moderate edema and early-stage epithelial shedding. Despite these findings, tissue architecture was not yet diffusely compromised, suggesting a subthreshold inflammatory activation at this exposure duration.

In contrast, the 30 min exposure group displayed severe histopathological alterations. Lung sections exhibited prominent vascular congestion, widespread alveolar wall thickening, diffuse edema, and extensive epithelial damage. Notably, bronchiolar epithelium appeared denuded in multiple fields, and the presence of mononuclear inflammatory infiltrates was markedly elevated. These findings strongly indicate a threshold beyond which the electric field disrupts pulmonary microcirculation and epithelial stability, facilitating acute inflammation and barrier compromise. Quantitative semi-scoring analysis substantiated these microscopic observations (Table 1, Figure 1).

Hyperemia scores were significantly higher in the 30 min group compared to all other groups, including the 5 and 15 min cohorts (*p* < 0.001). Similarly, edema severity sharply increased in the 30 min group, diverging markedly from the minimal findings in shorter exposure durations (*p* < 0.001). Infiltration of mononuclear cells followed a comparable trend, with the 30 min group exhibiting scores nearly sevenfold higher than the 5 min group.

The most striking difference was observed in epithelial loss. While minimal shedding was detected in the 5 and 15 min groups, the 30 min group displayed significantly elevated epithelial damage (*p* < 0.001). Post hoc comparisons confirmed the statistical significance of these findings between the 30 min group and all others. No significant differences were observed between the 5 and 15 min groups across all parameters (*p* > 0.999), reinforcing the concept of a cumulative threshold effect rather than a linear progression.

These results collectively demonstrate that prolonged exposure to a 10 kV/m electric field triggers a robust histopathological response in pulmonary tissue, primarily characterized by inflammation, epithelial barrier disruption, and vascular compromise. The absence of significant damage in lower exposure durations suggests the existence of an exposure-dependent tipping point, which may reflect a transition from adaptive to injurious cellular responses.

### 2.2. Immunohistochemical Evaluation of IL-1β Expression

Immunohistochemical analysis of IL-1β expression in lung tissues revealed a significant, exposure duration-dependent pro-inflammatory response induced by high-voltage electric field stimulation. In the control group, IL-1β immunoreactivity was virtually absent, with no detectable staining in bronchiolar, alveolar, or interstitial compartments. Similarly, the 1 min and 5 min exposure groups displayed only faint cytoplasmic staining localized to occasional alveolar epithelial cells and perivascular spaces, indicating a minimal basal activation of inflammatory cascades.

A modest increase in IL-1β immunoexpression was noted in the 15 min group, characterized by light brown cytoplasmic staining in scattered bronchiolar epithelial cells and peribronchial mononuclear elements. This focal elevation likely reflects a transitional inflammatory response, consistent with histopathological findings of low-grade infiltration and edema at this time point.

By contrast, a marked increase in IL-1β immunopositivity was observed in the 30 min exposure group. Strong cytoplasmic staining was diffusely distributed throughout the bronchial epithelium, alveolar septa, and perivascular inflammatory foci. In particular, the intensity and spatial distribution of IL-1β expression correlated well with areas of epithelial desquamation and mononuclear cell infiltration noted in corresponding H&E sections. The dense and homogeneous staining pattern strongly supports the activation of innate immune signaling in response to prolonged high-voltage exposure. Quantitative semi-scoring confirmed these histological observations (Table 2, Figure 2).

Mean IL-1β expression scores rose in the 5 min group compared to the 1 min group (*p* < 0.05). The 30 min group exhibited a sharp and statistically significant increase, which was highly elevated compared to all other groups (*p* < 0.001).

Post hoc analysis revealed no significant difference between the 5 min and 15 min groups (*p* > 0.999), but the 30 min group demonstrated a robust difference versus all others (*p* < 0.001). These findings suggest that while minimal inflammatory signaling may occur at short durations, a critical threshold of exposure is necessary to provoke IL-1β–mediated innate immune activation in lung tissue.

Overall, these results reinforce the hypothesis that high-voltage electric field exposure induces a sharp transition from subclinical to overt inflammation beyond a certain duration, with IL-1β acting as a key cytokine in this response. The strong correlation between IL-1β expression and histopathological injury in the 30 min group highlights its potential utility as a biomarker for electric field-induced pulmonary inflammation.

### 2.3. Immunohistochemical Evaluation of TNF-α Expression

The immunohistochemical expression pattern of TNF-α, a key pro-inflammatory cytokine, revealed a robust, exposure duration-dependent upregulation in lung tissue following high-voltage electric field application. In the control group, TNF-α staining was entirely absent, with no detectable immunoreactivity in bronchiolar epithelium, alveolar structures, or interstitial compartments, confirming the physiological quiescence of inflammatory signaling under baseline conditions.

In the 1 min and 5 min exposure groups, TNF-α immunoreactivity remained minimal and was limited to weak cytoplasmic staining in scattered bronchiolar epithelial cells. These findings suggest that short-term electric field exposure does not substantially trigger the NF-κB–mediated inflammatory cascade, nor does it induce overt cytokine release at the tissue level.

By the 15 min exposure point, a slight increase in TNF-α expression was observed. Mild cytoplasmic staining was present in peribronchial and perivascular regions, corresponding to focal areas of histopathological infiltration and edema. However, this moderate expression pattern remained localized and lacked the intensity or distribution necessary to indicate systemic inflammatory activation.

A striking elevation in TNF-α immunopositivity was observed in the 30 min group. Dense and diffuse cytoplasmic staining was clearly evident in bronchiolar and alveolar epithelial cells, as well as in infiltrating mononuclear cells within perivascular spaces. The spatial localization of TNF-α overlapped with histologically verified sites of epithelial loss and immune cell infiltration, strongly suggesting that TNF-α plays a central role in mediating the lung injury cascade triggered by prolonged electric field exposure.

Semi-quantitative scoring analysis revealed a substantial difference in TNF-α levels between the 30 min group and all other groups. While the mean expression scores remained low in the 1 to 15 min groups, a marked increase was documented in the 30 min group, with highly significant intergroup differences (*p* < 0.001). No significant differences were found between any of the shorter exposure groups (*p* > 0.999), indicating a binary-like transition into a pro-inflammatory state only after a certain threshold duration (Table 2, Figure 3).

These findings collectively confirm that prolonged high-voltage electric field exposure induces an acute inflammatory response in lung tissue, with TNF-α acting as a key molecular effector. The expression profile of TNF-α closely mirrors both histopathological injury and IL-1β/RANKL immunoreactivity, reinforcing its pivotal role in the amplification and maintenance of inflammation in this model.

### 2.4. Immunohistochemical Evaluation of RANKL Expression

Immunohistochemical analysis of RANKL expression in lung tissue revealed a significant and exposure-duration-dependent increase following high-voltage electric field stimulation. In the control group, no RANKL immunopositivity was detected in alveolar or bronchiolar regions, consistent with the physiological absence of active osteoimmunological signaling in normal pulmonary architecture.

Minimal cytoplasmic staining was observed in the 1 min and 5 min exposure groups, restricted to occasional bronchiolar epithelial cells and peribronchiolar interstitial zones. These findings likely represent low-level, non-specific activation of resident immune or epithelial cells in response to brief stimulation. Notably, these low-grade responses did not correlate with significant histopathological changes in corresponding H&E-stained sections, further supporting their limited biological impact.

A comparable immunoreactivity pattern was observed in the 15 min group, where RANKL staining remained focal and mild, with no expansion in staining intensity or distribution. This suggests that up to 15 min of exposure may fall within the adaptive threshold of pulmonary tissue, insufficient to elicit significant pro-resorptive or immunostimulatory effects.

In contrast, the 30 min exposure group exhibited a pronounced increase in RANKL expression. Strong cytoplasmic and membranous immunoreactivity was diffusely distributed across the bronchiolar epithelium, perivascular regions, and inflammatory cell clusters. This widespread upregulation of RANKL coincided with histopathological findings of epithelial loss, edema, and mononuclear infiltration, suggesting that RANKL may be actively involved in mediating electric field-induced pulmonary inflammation and tissue remodeling. Semi-quantitative scoring analysis supported these visual observations (Table 2, Figure 4).

While the control group and shorter exposure groups (1, 5, and 15 min) maintained consistently low expression scores, the 30 min group showed a substantial elevation in RANKL expression, which was statistically significant compared to all other groups (*p* < 0.001).

Post hoc analysis revealed no meaningful differences among the 1, 5, and 15 min groups (*p* > 0.999 for all comparisons), reinforcing the concept of threshold-dependent activation. The exclusive and robust upregulation of RANKL in the 30 min group points toward its role as a late-phase mediator of electric field-induced immune modulation in lung tissue.

Collectively, these findings indicate that prolonged exposure to high-voltage electric fields activates RANKL signaling in the lungs, potentially contributing to the amplification of local immune responses and tissue degradation pathways. The distinct elevation in RANKL immunoreactivity observed only after extended exposure supports its utility as a biomarker of electric field–associated pulmonary injury and epithelial remodeling.

## 3. Discussion

The present study demonstrates that exposure of lung tissue to a high-intensity (10 kV/m) electric field elicits a duration-dependent pathological response, characterized by progressive histological damage and marked upregulation of acute-phase inflammatory mediators. These findings bear significant clinical relevance, particularly in thoracic surgical and intensive care unit settings where patients are frequently subjected to high-intensity electromagnetic and electric fields during electrocautery, defibrillation, and advanced monitoring procedures. Although these tools are indispensable in modern practice, their potential to induce subclinical or overt pulmonary inflammation has been largely underestimated.

Our histopathological findings reveal that brief exposure durations (1–5 min) cause only minimal structural perturbations, with preserved alveolar architecture and limited vascular congestion. However, once exposure exceeds 15 min, a threshold appears to be crossed. The 30 min exposure group displayed severe histological disruption, including epithelial desquamation, diffuse edema, and dense mononuclear cell infiltration. These findings support the concept of a cumulative bioelectrical stress load, where lung tissue initially compensates for external stimuli but ultimately decompensates upon prolonged exposure.

This transition was paralleled by a distinct pattern of immunohistochemical activation. The acute-phase cytokines IL-1β, TNF-α, and RANKL remained at basal levels during the initial 15 min but increased sharply in the 30 min group. IL-1β, a prototypical early-phase cytokine produced by alveolar macrophages and epithelial cells, is a known amplifier of pulmonary inflammation, promoting neutrophil infiltration and vascular permeability [10]. The sudden rise in IL-1β in the 30 min group coincided spatially with epithelial erosion and inflammatory foci, suggesting that prolonged electric field exposure may directly trigger inflammasome activation or NF-κB–dependent signaling.

Similarly, TNF-α, a key effector in lung injury and acute respiratory distress syndrome (ARDS), exhibited a dramatic increase after 30 min, consistent with the role of electric field-induced oxidative stress in cytokine induction [11]. TNF-α is known to sensitize epithelial cells to apoptosis and disrupt tight junction integrity, both of which were reflected histologically in our study by alveolar wall damage and inflammatory infiltration.

The inclusion of RANKL adds a novel dimension to the inflammatory response observed. Traditionally linked to osteoclastogenesis, RANKL has recently been implicated in lung remodeling, immune cell recruitment, and epithelial–mesenchymal transition [12]. Its selective upregulation in the 30 min group points toward its involvement not only in chronic pulmonary diseases such as fibrosis but also in acute electric field-mediated tissue reprogramming. This finding positions RANKL as a potential immunomodulatory biomarker in lung injury secondary to environmental or procedural electric exposure.

Beyond its classical osteoimmunological role, recent studies have highlighted RANKL as a pivotal mediator of pulmonary inflammation and tissue remodeling. RANKL expression in alveolar epithelial and immune cells contributes to epithelial–mesenchymal transition, fibroblast activation, and recruitment of mononuclear cells during acute lung injury [7]. Moreover, RANKL-RANK signaling has been shown to enhance TNF-α and IL-1β production through NF-κB activation, creating a self-perpetuating inflammatory loop that aggravates tissue damage [12]. These findings suggest that the strong RANKL upregulation observed in the 30 min exposure group may reflect a transition from acute cytokine signaling toward structural remodeling within the lung parenchyma.

Our findings align with prior experimental models in other organ systems, such as the kidney and brain, where electric field exposure has shown biphasic effects—initially promoting adaptive antioxidant responses, but, upon exceeding a critical exposure threshold, precipitating inflammation and apoptosis [13,14]. The lung, with its extensive surface area and thin epithelial barrier, may be particularly vulnerable to such perturbations, especially in ventilated or surgically manipulated patients.

Importantly, the histopathological and molecular findings in this study mirror real-world clinical exposure scenarios. Electrocautery units can generate local electric fields exceeding 10 kV/m, and intraoperative exposure times may approach or exceed 30 min in complex thoracic resections [15]. Moreover, patients with prolonged ICU stays, especially those on non-shielded monitors and devices, may be subject to low-grade chronic electric field exposure, the implications of which remain unknown [16].

The observed histopathological and immunohistochemical findings suggest that high-intensity electric field exposure initiates a cascade of redox-sensitive inflammatory events. Although direct measurements of oxidative stress markers were not performed due to budget limitations, the abrupt increase in IL-1β, TNF-α, and RANKL expression at the 30 min exposure point strongly indicates the activation of upstream redox and inflammasome-related pathways. Previous studies have demonstrated that prolonged electric field exposure can elevate intracellular ROS and lipid peroxidation, leading to the activation of the NOD-like receptor protein 3 (NLRP3) inflammasome and phosphorylation of NF-κB p65 subunits [13,14]. Similar mechanisms have been reported in hepatic and neural tissues following electromagnetic or electric field stress, where reactive oxygen species (ROS) accumulation precedes the release of pro-inflammatory cytokines [10,11]. Therefore, it is plausible that the 30 min exposure in the current study surpassed the adaptive antioxidant threshold of lung tissue, triggering NF-κB–dependent transcription of IL-1β and TNF-α, and consequently enhancing RANKL-mediated epithelial–mesenchymal signaling. The synchronous rise in these mediators supports an ROS–inflammasome–cytokine axis as the underlying mechanism of electric field-induced pulmonary injury.

From a clinical perspective, our findings suggest that while short-term electric field exposure may be tolerable or even biologically inert, extended exposure durations can trigger robust pulmonary inflammation, likely through the activation of innate immune pathways and cytokine cascades. This raises important considerations for the design of operating room protocols, shielding of ICU equipment, and timing of energy-based device usage, particularly in patients with preexisting lung disease or compromised pulmonary function.

High-intensity electric fields can theoretically generate Joule heating; however, in our setup, the parallel-plate design and open airflow prevented measurable temperature elevation. Continuous thermometric recordings confirmed a constant 22 ± 2 °C within the chamber, thereby excluding heat-induced tissue injury.

Similarly, although physical restraint can elevate circulating corticosterone and modulate immune responses, all experimental and control animals experienced identical restraint conditions and environmental stimuli. Therefore, the differential lung injury and cytokine upregulation observed in the 30 min group cannot be attributed to stress alone but rather to cumulative electric field exposure. This standardized approach minimizes confounding factors and strengthens the causal interpretation of electric field-induced inflammation. To improve visual clarity, all graphs have been reformatted as single bar plots with individual data points (mean ± SD), and all redundant overlays were removed.

This study is not without limitations. First, the model utilized a uniform, controlled electric field that may not fully replicate the heterogeneous field distributions seen in clinical practice. Second, the absence of direct ROS measurement or NF-κB activation limits the mechanistic depth of the inflammatory pathway elucidation. Molecular assays such as Western blotting for NF-κB p65, NLRP3, and caspase-1 activation, as well as biochemical measurements of ROS, MDA, or SOD, could not be conducted. Nevertheless, the consistent pattern of histopathological injury and cytokine upregulation provides strong indirect evidence of redox-driven inflammatory activation. Future studies will incorporate these analyses and complementary in vitro assays using lung epithelial and macrophage cultures to directly validate this mechanistic pathway. Another limitation of the present study is that only female Wistar rats were used. Female animals were selected to minimize aggressive behavior and reduce hormonal variability during housing and exposure procedures. However, sex-related differences in pulmonary inflammatory responses are well recognized; therefore, future studies should include both sexes to validate the generalizability of these findings.

## 4. Materials and Methods

### 4.1. Animals

A total of thirty-five adult female Wistar albino rats, aged 12–14 weeks and weighing 250 to 300 g, were procured from the Suleyman Demirel University Experimental Animal Center. Throughout the study, the animals were maintained in individually ventilated cages under standardized laboratory conditions, including a stable ambient temperature of 22 ± 2 °C, a relative humidity range of 55–65%, and a 12 h light/dark cycle. Standard rodent chow and water were provided ad libitum.

### 4.2. Study Design

After completing the acclimatization phase, the rats were randomly divided into five experimental groups, each consisting of eight animals (*n* = 7), according to the duration of electric field exposure planned for the study:

**Group I (Sham Control):** Animals were placed in the exposure apparatus without activating the electric field, serving to control for environmental influences.**Group II (1-Minute Exposure):** Exposed once to a 10 kV/m electric field for a duration of 1 min.**Group III (5-Minute Exposure):** Subjected to the same 10 kV/m electric field for 5 min.**Group IV (15-Minute Exposure):** Continuously exposed to the electric field for 15 min.**Group V (30-Minute Exposure):** Received prolonged exposure of 30 min under identical field conditions [8].

Electric field application was performed using a previously standardized parallel-plate configuration, producing a uniform intensity of 10 kV/m. Animals were placed in a transparent, non-conductive acrylic chamber positioned between the plates to ensure even exposure and to prevent any direct contact with electrical components. No sedatives or anesthetics were administered during the procedure; however, physical restraint was applied gently to minimize movement and maintain consistent exposure parameters. In addition, all animals, including those in the sham control group, were subjected to identical handling and restraint conditions for the same duration as their respective exposure groups, with the electric field generator turned off. This design ensured that any physiological stress resulting from physical restraint was equally distributed across all groups and did not influence between-group differences.

The exposure system consisted of two parallel, stainless steel-coated plates (100 × 50 cm) positioned 70 cm apart, generating a homogeneous field verified at 10 ± 0.2 kV/m using a calibrated high-voltage probe (Testo 925, Titisee-Neustadt, Germany). The parallel-plate geometry and grounded shielding minimized field distortion. To rule out possible thermal artifacts, the temperature inside the acrylic exposure chamber was continuously monitored using a digital thermoprobe (Testo 925, Germany) positioned at the center of the animal compartment throughout all exposure sessions. The recorded temperature remained stable at 22 ± 2 °C for the entire duration of 1–30 min exposures, confirming that no measurable heat generation occurred within the chamber.

Control box can produce 7 kV potential from the wall outlet (220 V to 7 kV). Also, this control device can produce a stable supply voltage. E = V/d
where V is the potential difference between plates, d is the distance between plates, and E is the produced electric field.E = 7/0.7E = 10 kV/m

In other words, the daily exposure level of electric field was produced as 10 kV/m. According to WHO guidelines, in this power frequency range, 10 kV/m is the limit for public exposure [9]. A schematic of the uniform field setup is presented in Figure 5.

Post-exposure, each rat was monitored for behavioral changes, alterations in vital function, and signs of distress. At the end of their designated exposure intervals, the animals were anesthetized intraperitoneally with 90 mg/kg ketamine and 8–10 mg/kg xylazine, subsequently sacrificed, and lung tissues were harvested for histopathological and immunohistochemical analyses.

All experimental procedures adhered to the ethical guidelines outlined in the *NIH Guide for the Care and Use of Laboratory Animals*, and ethical approval was obtained from the Süleyman Demirel University Local Ethics Committee for Animal Experiments (Approval No: [08/618]).

### 4.3. Histopathological Analysis

Lung tissue samples were collected during necropsy and fixed in 10% buffered formalin for two days. Following fixation, the samples were routinely processed and embedded in paraffin wax using a fully automated tissue processor (Leica ASP300S; Leica Microsystem, Nussloch, Germany). Five-micron sections were then obtained from the paraffin blocks using a fully automated Leica RM 2155 rotary microtome (Leica Microsystem, Nussloch, Germany). The sections were stained with hematoxylin–eosin (H&E), mounted with coverslips, and examined under a light microscope.

Histopathological lesions were evaluated using a severity-based scoring system (0–3) for hyperemia, edema, inflammatory cell infiltration, and epithelial cell loss, following the criteria established by Ozmen et al. [17] (Table 3).

### 4.4. Immunohistochemical Analysis

Three series of sections obtained from each paraffin block were mounted on poly-L-lysine–coated slides and immunohistochemically stained for the expression of IL-1β, TNF-α, and RANKL using recombinant antibodies: IL-1β antibody (Recombinant Anti-IL-1 beta antibody [RM1009] (ab283818)), TNF-α (Anti-TNF alpha recombinant antibody [RM1005], ab307164), and RANKL (Anti-sRANKL antibody (ab62516)), following the manufacturer’s instructions. Each primary antibody was diluted at 1:100. Immunohistochemistry was performed using a streptavidin–alkaline phosphatase conjugate and a biotinylated secondary antibody after overnight incubation with primary antibodies. The EXPOSE Mouse and Rabbit Specific HRP/DAB Detection IHC kit (ab80436) was used for secondary antibody detection, with diaminobenzidine (DAB) as the chromogen. All primary and secondary antibodies were supplied by Abcam (Cambridge, UK). As a negative control, antigen dilution solution was applied instead of primary antibodies. A specialized pathologist from a different university conducted all analyses on blinded samples.

Immunopositive cells were quantified at 40× magnification by counting the percentage of stained cells in 10 randomly selected fields per slide across all groups. Image analysis was performed using ImageJ software (National Institutes of Health, Bethesda, MD, USA, version 1.48). Prior to counting, images were cropped, divided into color channels, and artifacts were removed. Cells within the regions of interest were selected using a selection tool, and only those displaying strong red staining were considered positive. Microphotographs were captured using the Database Manual Cell Sens Life Science Imaging Software System (Olympus Co., Tokyo, Japan) (https://evidentscientific.com/en/products/software/cellsens, accessed on 17 September 2025) [17].

### 4.5. Statistical Analysis

The normality of data distribution was first assessed using the Shapiro–Wilk test. As the data were normally distributed (*p* > 0.05), group comparisons were conducted using one-way ANOVA. Post hoc analysis was performed using Tukey’s test to determine pairwise differences between groups. All statistical analyses were performed using GraphPad Prism 10.0.1, and results are presented as mean ± SD. A *p*-value of less than 0.05 was considered statistically significant.

## 5. Conclusions

High-intensity electric field exposure induces a distinct and duration-dependent inflammatory phenotype in lung tissue, characterized by epithelial injury and upregulation of acute-phase reactants such as IL-1β, TNF-α, and RANKL. These findings underscore the need for increased awareness and regulation of electric field exposure in thoracic surgical and critical care environments, with particular attention to exposure duration and cumulative energy load. The integration of inflammatory biomarker screening in perioperative monitoring may offer a future avenue for individualized risk stratification and lung protection strategies. Collectively, even in the absence of direct molecular quantification, the convergent histological and cytokine findings delineate a plausible oxidative–inflammatory cascade underlying electric field-induced pulmonary injury.

## Figures and Tables

**Figure 1 ijms-26-10125-f001:**
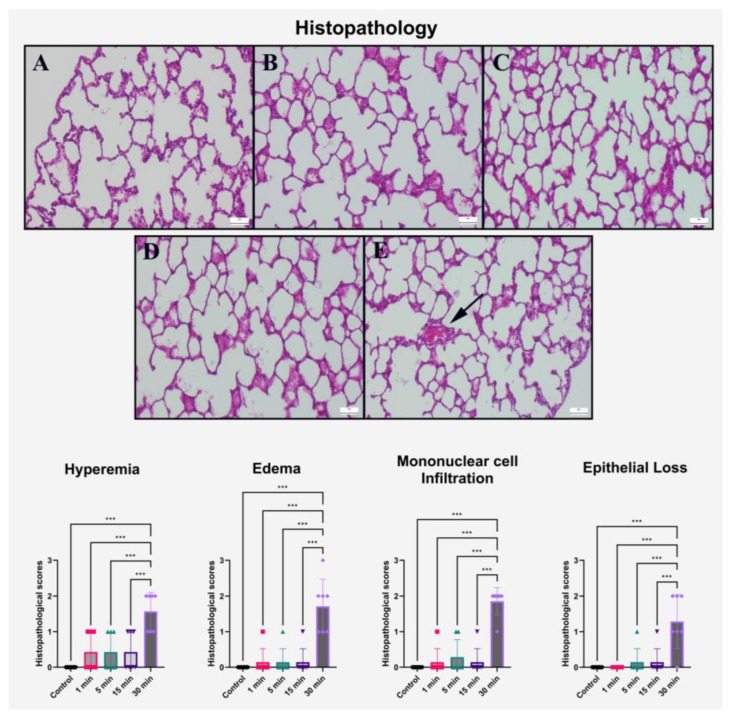
Histopathological evaluation of rat lung tissues stained with hematoxylin and eosin (H&E) following graded-duration exposure to a 10 kV/m electric field. (**A**–**E**) Representative sections from control, 1 min, 5 min, 15 min, and 30 min groups, respectively. Control tissues exhibited preserved alveolar and bronchiolar structure. Mild vascular congestion was observed at 1 and 5 min. The 15 min group demonstrated focal inflammatory infiltration and early epithelial compromise. The 30 min group revealed extensive epithelial desquamation, dense mononuclear infiltration (black arrow), and interstitial edema. Scale bar = 50 μm. Bar graphs represent mean ± SD (*n* = 7) with individual data points displayed. Statistical analysis was performed using one-way ANOVA followed by Tukey’s post hoc test. *** *p* < 0.001.

**Figure 2 ijms-26-10125-f002:**
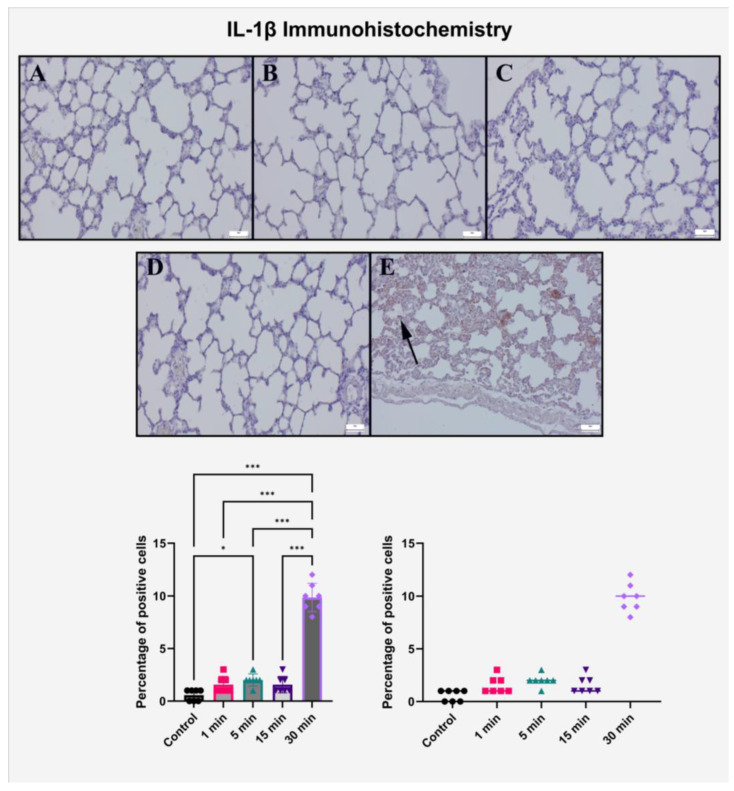
Immunohistochemical evaluation of IL-1β expression in rat lung tissue after graded high-intensity electric field exposure. (**A**–**E**) 1 min, 5 min, 15 min, and 30 min groups, respectively. IL-1β expression was absent in control and weakly positive in 1–15 min groups, with localized cytoplasmic staining. In the 30 min group, strong cytoplasmic IL-1β positivity was observed diffusely across bronchial and alveolar epithelium, particularly in perivascular infiltrates (black arrow). Scale bar = 50 μm. Bar graphs represent mean ± SD (*n* = 7) with individual data points displayed. Statistical analysis was performed using one-way ANOVA followed by Tukey’s post hoc test. * *p* < 0.05, *** *p* < 0.001 compared to all other groups. All immunohistochemical scores were calculated as the percentage of positive cells.

**Figure 3 ijms-26-10125-f003:**
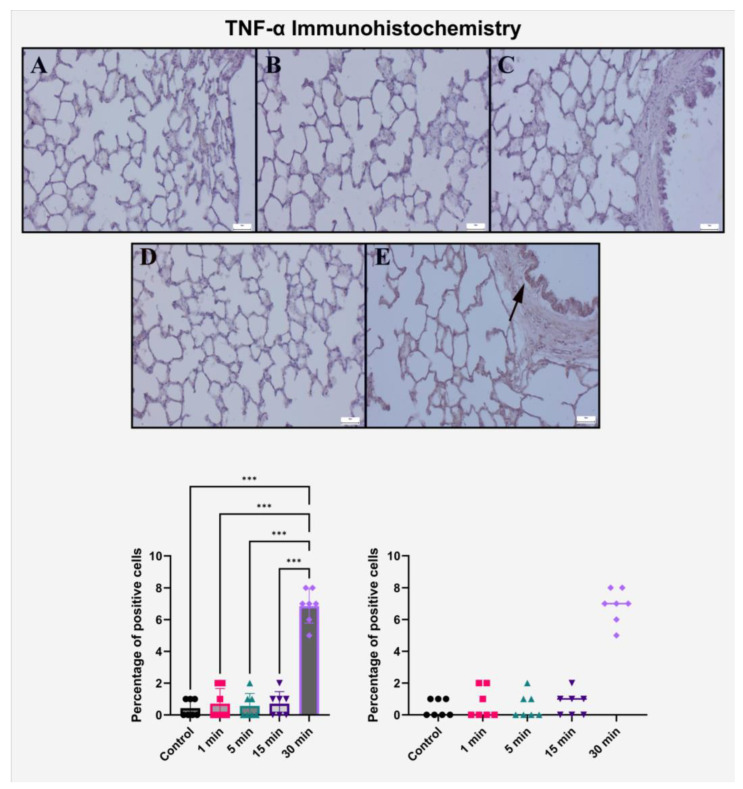
Immunohistochemical expression of TNF-α in lung tissue after graded exposure to a 10 kV/m electric field. (**A**–**E**) 1 min, 5 min, 15 min, and 30 min groups, respectively. TNF-α immunoreactivity was absent in the control group and remained weak in 1–15 min groups. The 30 min group exhibited intense, diffuse cytoplasmic staining (black arrow) in bronchiolar epithelium and infiltrating mononuclear cells. Scale bar = 50 μm. Bar graphs represent mean ± SD (*n* = 7) with individual data points displayed. Statistical analysis was performed using one-way ANOVA followed by Tukey’s post hoc test. *** *p* < 0.001 compared to all other groups. All immunohistochemical scores were calculated as the percentage of positive cells.

**Figure 4 ijms-26-10125-f004:**
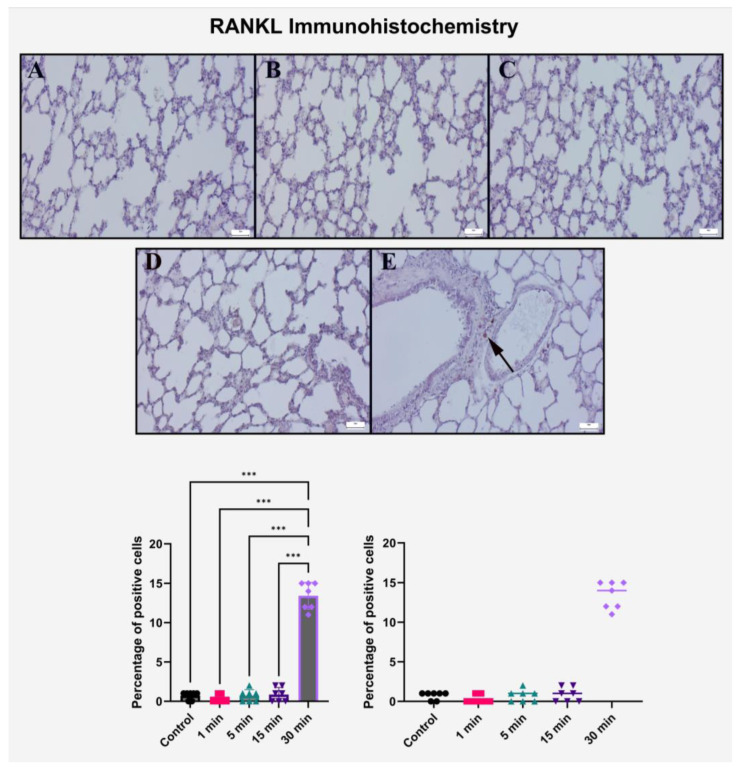
Immunohistochemical analysis of RANKL expression in rat lung tissue following increasing durations of 10 kV/m electric field exposure. (**A**–**E**) 1 min, 5 min, 15 min, and 30 min groups, respectively. RANKL expression was negligible in control and remained weakly positive in 1–15 min groups, with limited focal staining in epithelial cells. The 30 min group demonstrated diffuse and strong RANKL positivity (black arrow), particularly in bronchiolar epithelium and perivascular zones. Scale bar = 50 μm. Bar graphs represent mean ± SD (*n* = 7) with individual data points displayed. Statistical analysis was performed using one-way ANOVA followed by Tukey’s post hoc test. *** *p* < 0.001 compared to all other groups. All immunohistochemical scores were calculated as the percentage of positive cells.

**Figure 5 ijms-26-10125-f005:**
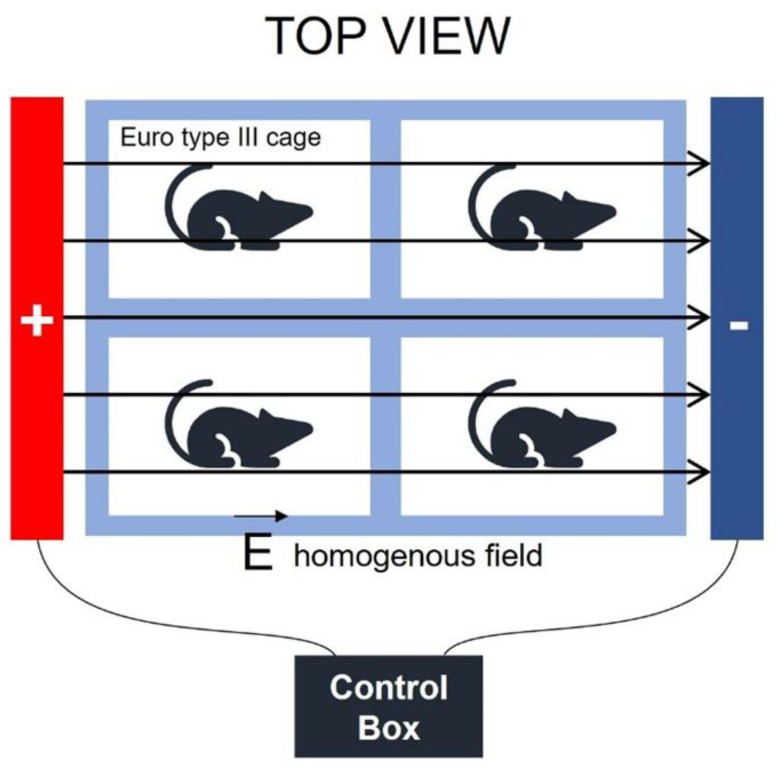
Electric field application unit.

**Table 1 ijms-26-10125-t001:** Histopathological scores of lung tissue.

Group	Hyperemia (Mean ± SD)	Edema (Mean ± SD)	Infiltration (Mean ± SD)	Epithelial Loss (Mean ± SD)
**Control**	0.00 ± 0.00	0.00 ± 0.00	0.00 ± 0.00	0.00 ± 0.00
**1 min**	0.43 ± 0.53	0.14 ± 0.38	0.14 ± 0.38	0.14 ± 0.38
**5 min**	0.43 ± 0.53	0.14 ± 0.38	0.29 ± 0.49	0.14 ± 0.38
**15 min**	0.43 ± 0.53	0.14 ± 0.38	0.14 ± 0.38	0.29 ± 0.38
**30 min**	1.57 ± 0.53	1.71 ± 0.76	1.86 ± 0.38	1.29 ± 0.76

Data represent the mean ± standard deviation (SD) for histopathological evaluation across groups.

**Table 2 ijms-26-10125-t002:** Summary of scores and adjusted *p*-values for immunohistochemical evaluations of lung tissue.

Group	IL-1β (Mean ± SD)	RANKL (Mean ± SD)	TNF-α (Mean ± SD)
**Control**	0.57 ± 0.53	0.71 ± 0.49	0.43 ± 0.53
**1 min**	1.57 ± 0.79	0.29 ± 0.49	0.71 ± 0.95
**5 min**	2.00 ± 0.58	0.71 ± 0.76	0.57 ± 0.79
**15 min**	1.57 ± 0.79	0.86 ± 0.90	0.71 ± 0.76
**30 min**	9.86 ± 1.35	13.43 ± 1.72	6.86 ± 1.07
**Marker**	**Comparison**	**Adjusted *p*-Value**	**Significance**
**IL-1β**	Control vs. 1 min	0.213	ns
**IL-1β**	Control vs. 5 min	0.030	*
**IL-1β**	Control vs. 15 min	0.213	ns
**IL-1β**	Control vs. 30 min	<0.001	***
**IL-1β**	1 min vs. 5 min	0.880	ns
**IL-1β**	1 min vs. 15 min	>0.999	ns
**IL-1β**	1 min vs. 30 min	<0.001	***
**IL-1β**	5 min vs. 15 min	0.880	ns
**IL-1β**	5 min vs. 30 min	<0.001	***
**IL-1β**	15 min vs. 30 min	<0.001	***
**RANKL**	Control vs. 1 min	0.923	ns
**RANKL**	Control vs. 5 min	>0.999	ns
**RANKL**	Control vs. 15 min	0.999	ns
**RANKL**	Control vs. 30 min	<0.001	***
**RANKL**	1 min vs. 5 min	0.923	ns
**RANKL**	1 min vs. 15 min	0.810	ns
**RANKL**	1 min vs. 30 min	<0.001	***
**RANKL**	5 min vs. 15 min	0.999	ns
**RANKL**	5 min vs. 30 min	<0.001	***
**RANKL**	15 min vs. 30 min	<0.001	***
**TNF-α**	Control vs. 1 min	0.968	ns
**TNF-α**	Control vs. 5 min	0.998	ns
**TNF-α**	Control vs. 15 min	0.968	ns
**TNF-α**	Control vs. 30 min	<0.001	***
**TNF-α**	1 min vs. 5 min	0.998	ns
**TNF-α**	1 min vs. 15 min	>0.999	ns
**TNF-α**	1 min vs. 30 min	<0.001	***
**TNF-α**	5 min vs. 15 min	0.998	ns
**TNF-α**	5 min vs. 30 min	<0.001	***
**TNF-α**	15 min vs. 30 min	<0.001	***

IL-1β, RANKL, and TNF-α in lung tissue following graded electric field exposure. Values are presented as mean ± SD. Adjusted *p*-values derived from Tukey’s post hoc multiple comparison test for pairwise group comparisons in immunohistochemical parameters. * *p* < 0.05, *** *p* < 0.001, ns: not significant.

**Table 3 ijms-26-10125-t003:** Histopathological score criterion for lungs.

Score	Hyperemia	Edema	Infiltrations	Epithelial Loss
**0**	Normal thin alveolar septae	No edema	Less than 5 in the fields	None
**1**	Slight hyperemic alveolar septae (in less than 1/3 of the fields)	Slight edema (in less than 1/3 of the fields)	5 to 10 in the fields	Slight loss (in less than 1/3 of the fields)
**2**	Moderate hyperemic alveolar septae (in 1/3 to 2/3 of the fields)	Moderate edema (in 1/3 to 2/3 of the fields)	10 to 20 in the fields	Moderate loss (in 1/3 to 2/3 of the fields)
**3**	Severe hyperemic alveolar septae (in greater than 2/3 of the fields)	Severe edema (in greater than 2/3 of the fields)	More than 20 in the fields	Severe edema (in greater than 2/3 of the fields)

Examinations performed with high-power field (HPF) (10 fields of view at ×400 magnification).

## Data Availability

The data of the study results are available from the corresponding author upon request.

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
