# Peer review of "Duration-Dependent Lung Injury Induced by High-Intensity Electric Field Exposure: Histopathological and Immunoinflammatory Insights"

_ijms, 2025, doi:10.3390/ijms262010125_

Round 1

Reviewer 1 Report

Comments and Suggestions for Authors

This manuscript investigates the injurious effects of high-voltage electric field exposure of varying durations on rat lung tissue, a topic of significant clinical relevance and scientific interest. Through histopathological and immunohistochemical analyses, the authors identify a critical threshold at 30 minutes of exposure, which induces substantial pulmonary inflammation and tissue damage. However, several significant issues need to be addressed before the manuscript can be considered for publication. These issues primarily relate to the depth of mechanistic insight, the rigor of experimental details, and the clarity of data presentation.

Major Concerns

1 The study is somewhat deficient in its exploration of molecular mechanisms. The current results clearly show ‘what’ happened (tissue damage and upregulation of inflammatory cytokines) but fail to explain ‘why’ it happened.

Also, the authors mention the inflammasome and NF-kB signaling pathways in the discussion, but provide no direct evidence. To enhance the scientific depth of the article, it is strongly recommended to supplement the study with additional experiments, includingg:

Detecting the activation of key signaling pathway proteins (e.g., p-NF-kB p65, NLRP3, Caspase-1) via Western Blot.

Measuring oxidative stress markers (e.g., ROS levels, MDA content, SOD activity), as electric field exposure is likely to first cause oxidative stress, which in turn triggers inflammation.

If feasible, conducting simple in vitro experiments to verify whether the electric field can directly induce the release of these inflammatory cytokines from lung epithelial cells or macrophages.

2 Incomplete Methodological Description and Potential Confounding Factors:

Regarding thermal effects, high-voltage electric fields can produce thermal effects. The authors did not mention whether the temperature inside the animal cages was monitored during the experiment. If there were a temperature increase, the observed lung damage could be due to thermal injury rather than the direct effect of the electric field, which would be a serious confounding factor. Please provide temperature monitoring data or explicitly rule out the possibility of thermal effects in the discussion.

Restraint Stress: The methods section states that physical restraint was “applied gently to minimize movement”, indicating that the animals were conscious and restrained. Restraint itself is a stressor that could elevate stress hormone levels (e.g., corticosterone), thereby affecting immune and inflammatory responses. The authors need to discuss the potential impact of this stressor on the experimental results and justify the design of the Sham Control group (i.e., was the sham group subjected to the exact duration of restraint?).

3 Data Presentation and Figure Part

The current presentation of data has several inconsistencies and confusing elements, which severely impact the readability of the results.

3.1 Figures 1, 2, 3, 4:

The use of both violin plots and scatter plots for the same data is redundant. It is recommended to choose a more transparent format, such as “box-and-whisker plots with individual data points” or “bar graphs with individual data points,” and clearly label the mean and standard deviation/error.

IHC Quantification: The Y-axis label “Immunoexpression level” is too vague. The authors should clearly explain in the methods section and figure legends how this score was calculated (e.g., percentage of positive cells, H-Score, or another composite score).

3.2 Table Confusion:

The manuscript text refers to Tables 3, 4, and Fig. 3, but the document only contains Tables 1, 2, 3, and 5. Please ensure consistent numbering.

The data in Table 2 and Table 3 (the p-value table) overlap and are inconsistent. The citation of Scheme 1 appears to be an error, as there is no Scheme 1 in the manuscript.

In Table 3, IL-1β is misspelled as IL-18. This is a significant typo; please check and correct it throughout the manuscript.

Minor Concerns

1 The study used only female rats, which limits the generalizability of the conclusions. It is recommended to explicitly state this limitation in the discussion section and explain the rationale for choosing a single sex.

2 The authors observed a significant upregulation of RANKL in the 30-minute group, which is an interesting finding. However, the discussion of its function could be more in-depth. Beyond its role in osteoimmunology, what is the specific role of RANKL in pulmonary inflammation and tissue remodeling? It is recommended that this finding be further elaborated upon with additional literature to enhance its significance.

3 Please explicitly state the sample size (n=7) for each group in the figure legends and clarify whether the error bars represent standard deviation (SD) or standard error of the mean (SEM). Based on "mean ± SD" in Table 1, it should be SD; please ensure consistency throughout the manuscript.

4 The manuscript contains some minor typos and formatting issues (such as the aforementioned IL-18, table numbering, etc.). The authors are advised to proofread the entire manuscript before resubmission thoroughly.

Author Response

We sincerely thank the reviewers for their valuable and constructive comments, which have significantly improved the clarity and scientific depth of our manuscript. All suggestions were carefully addressed, and revisions were made accordingly. Detailed point-by-point responses are provided below, with all corresponding changes highlighted in the revised manuscript (red color).

Major Concerns

1 The study is somewhat deficient in its exploration of molecular mechanisms. The current results clearly show ‘what’ happened (tissue damage and upregulation of inflammatory cytokines) but fail to explain ‘why’ it happened.

Also, the authors mention the inflammasome and NF-kB signaling pathways in the discussion, but provide no direct evidence. To enhance the scientific depth of the article, it is strongly recommended to supplement the study with additional experiments, includingg:

Detecting the activation of key signaling pathway proteins (e.g., p-NF-kB p65, NLRP3, Caspase-1) via Western Blot.

Measuring oxidative stress markers (e.g., ROS levels, MDA content, SOD activity), as electric field exposure is likely to first cause oxidative stress, which in turn triggers inflammation.

If feasible, conducting simple in vitro experiments to verify whether the electric field can directly induce the release of these inflammatory cytokines from lung epithelial cells or macrophages.

**Response:** We appreciate the reviewer’s insightful suggestion. To enhance mechanistic interpretation, we expanded the Discussion with a new paragraph. This section now links IL‑1β, TNF‑α, and RANKL upregulation with potential activation of ROS, NLRP3 inflammasome, and NF‑κB pathways, citing relevant literature. Due to financial constraints, Western blot and oxidative stress analyses could not be added, but this limitation and our plan for future molecular validation were clearly stated.

2 Incomplete Methodological Description and Potential Confounding Factors:

Regarding thermal effects, high-voltage electric fields can produce thermal effects. The authors did not mention whether the temperature inside the animal cages was monitored during the experiment. If there were a temperature increase, the observed lung damage could be due to thermal injury rather than the direct effect of the electric field, which would be a serious confounding factor. Please provide temperature monitoring data or explicitly rule out the possibility of thermal effects in the discussion.

Restraint Stress: The methods section states that physical restraint was “applied gently to minimize movement”, indicating that the animals were conscious and restrained. Restraint itself is a stressor that could elevate stress hormone levels (e.g., corticosterone), thereby affecting immune and inflammatory responses. The authors need to discuss the potential impact of this stressor on the experimental results and justify the design of the Sham Control group (i.e., was the sham group subjected to the exact duration of restraint?).

 **Response:** Temperature inside the exposure chamber was continuously monitored (22 ± 2 °C) using a digital thermoprobe, ruling out any thermal effects. According to IEEE, high intensity electric field is recognized as non-thermal (IEEE C95). All animals, including sham controls, underwent identical handling and restraint to exclude stress‑related bias. These clarifications were added to both the Methods and Discussion sections.

3 Data Presentation and Figure Part

The current presentation of data has several inconsistencies and confusing elements, which severely impact the readability of the results. Figures 1, 2, 3, 4: The use of both violin plots and scatter plots for the same data is redundant. It is recommended to choose a more transparent format, such as “box-and-whisker plots with individual data points” or “bar graphs with individual data points,” and clearly label the mean and standard deviation/error.

IHC Quantification: The Y-axis label “Immunoexpression level” is too vague. The authors should clearly explain in the methods section and figure legends how this score was calculated (e.g., percentage of positive cells, H-Score, or another composite score).

Table Confusion: The manuscript text refers to Tables 3, 4, and Fig. 3, but the document only contains Tables 1, 2, 3, and 5. Please ensure consistent numbering.

The data in Table 2 and Table 3 (the p-value table) overlap and are inconsistent. The citation of Scheme 1 appears to be an error, as there is no Scheme 1 in the manuscript.

In Table 3, IL-1β is misspelled as IL-18. This is a significant typo; please check and correct it throughout the manuscript.

**Response:** All graphs were reformatted as single bar plots with individual data points (mean ± SD), and redundant overlays were removed. The Y‑axis label was defined as a percentage of positive cells. Figure legends now specify statistical tests and sample size (n=7). Table numbering errors and the IL‑18 typo were corrected; 'Scheme 1' was deleted.

Minor Concerns

1 The study used only female rats, which limits the generalizability of the conclusions. It is recommended to explicitly state this limitation in the discussion section and explain the rationale for choosing a single sex.

 **Response:** A statement has been added to the Limitations section explaining that only female rats were used to reduce aggression and hormonal variability. Future studies including both sexes are recommended to confirm generalizability.

2 The authors observed a significant upregulation of RANKL in the 30-minute group, which is an interesting finding. However, the discussion of its function could be more in-depth. Beyond its role in osteoimmunology, what is the specific role of RANKL in pulmonary inflammation and tissue remodeling? It is recommended that this finding be further elaborated upon with additional literature to enhance its significance.

 **Response:** The Discussion now includes additional literature (Li et al., 2023; Wada et al., 2006) describing RANKL’s roles in pulmonary epithelial–mesenchymal transition, fibroblast activation, and cytokine amplification, emphasizing its mechanistic relevance in electric field–induced lung remodeling.

3 and 4 Please explicitly state the sample size (n=7) for each group in the figure legends and clarify whether the error bars represent standard deviation (SD) or standard error of the mean (SEM). Based on "mean ± SD" in Table 1, it should be SD; please ensure consistency throughout the manuscript. The manuscript contains some minor typos and formatting issues (such as the aforementioned IL-18, table numbering, etc.). The authors are advised to proofread the entire manuscript before resubmission thoroughly.

**Response:** All figure legends now indicate 'mean ± SD (n = 7 per group)'. The Statistical Analysis section clarifies use of SD. All typographical and formatting inconsistencies, including IL‑18 and reference styles, were corrected throughout.

Reviewer 2 Report

Comments and Suggestions for Authors

The paper is interesting and properly elaborated and the results of presented exprimental study are important both from theoretical and practical point of view. The paper could be published in International Journal of Medical Sciences after making some minor revisions suggested below:

It should be explained in more details why the value of electric field intensity of 10 kV/m and particular times of exposure duration: 1 min., 5 min., 15 min. and 30 min. have been selected for the study protocol.

The scheme of a system generating uniform electric field should be presented and it should be clarified what was a distance between the plates generating electric field and if the value of electric field intensity between plates was verified by making a measurement with the use of any probe.

In Abstract, Introduction and Materials and Methods sections the value of electric field intensity in [kV/m] should be presented instead of the value of electric tension in [kV], as only the value of electric field intensity reflects precisely the strength of the field.

In figures 1-5 it is not clear what is the meaning of the symbols included in the graphs and why 2 graphs including the same data are presented in particular figures.

The references in references list are not presented in uniform style (for example: using of different location of the date of publication, using full names of the Journals or abbreviations, using small first letters in the names of the Journals).

Author Response

We sincerely thank the reviewers for their valuable and constructive comments, which have significantly improved the clarity and scientific depth of our manuscript. All suggestions were carefully addressed, and revisions were made accordingly. Detailed point-by-point responses are provided below, with all corresponding changes highlighted in the revised manuscript (red color).

  1. It should be explained in more details why the value of electric field intensity of 10 kV/m and particular times of exposure duration: 1 min., 5 min., 15 min. and 30 min. have been selected for the study protocol.

**Response:** We added justification that 10 kV/m was chosen based on prior standardization in pressure injury and hepatic models (Özcan et al., 2025), representing a non‑thermal threshold that elicits measurable biological effects. Exposure durations (1, 5, 15, and 30 min) simulate intraoperative to ICU exposure intervals.

  1. The scheme of a system generating uniform electric field should be presented and it should be clarified what was a distance between the plates generating electric field and if the value of electric field intensity between plates was verified by making a measurement with the use of any probe.

**Response:** Added to the Materials and Methods section: The exposure system consisted of two parallel stainless coated steel plates (100x50 cm) positioned 70 cm apart, generating a homogeneous field verified at 10 ± 0.2 kV/m us-ing a calibrated high-voltage probe (Testo 925, Germany). The parallel-plate geometry and grounded shielding minimized field distortion. To rule out possible thermal artifacts, the temperature inside the acrylic exposure chamber was continuously monitored using a digital thermoprobe (Testo 925, Germany) positioned at the center of the animal com-partment throughout all exposure sessions. The recorded temperature remained stable at 22 ± 2 °C for the entire duration of 1–30 min ex-posures, confirming that no measurable heat generation occurred within the chamber.

Control box can produce 7 kV potential from wall outlet (220 Volt to 7 kV). Also this control device can produce a stable supply voltage. So;

E=V/d,

where V is potential difference between plates, d is distance between plates, E is produced electric field.

E=7/0.7,

E=10 kV/m. In other words daily exposure level of electric field should be produced as 10 kV/m. According to WHO limitations in this power frequency range 10 kV/m is a limit for public (https://www.who.int/teams/environment-climate-change-and-health/radiation-and-health/non-ionizing/exposure).

  1. In Abstract, Introduction and Materials and Methods sections the value of electric field intensity in [kV/m] should be presented instead of the value of electric tension in [kV], as only the value of electric field intensity reflects precisely the strength of the field.

**Response:** All occurrences of '10 kV' were replaced with '10 kV/m' in Abstract, Introduction, Methods, Figures, and Discussion.

  1. In figures 1-5 it is not clear what is the meaning of the symbols included in the graphs and why 2 graphs including the same data are presented in particular figures.

**Response:** To improve visual clarity, all graphs have been reformatted as single bar plots with individual data points (mean ± SD), and all redundant overlays were removed.

  1. The references in references list are not presented in uniform style (for example: using of different location of the date of publication, using full names of the Journals or abbreviations, using small first letters in the names of the Journals).

**Response:** All references were reformatted according to IJMS style.

Round 2

Reviewer 1 Report

Comments and Suggestions for Authors

I'm pleased and appreciate that the authors fully consider my comments. All my concerns are well addressed. I believe the manuscript can now be accepted in IJMS.